# Temperature and Light Spectrum Differently Affect Growth, Morphology, and Leaf Mineral Content of Two Indoor-Grown Leafy Vegetables

**Yun Kong †, Joseph Masabni and Genhua Niu ***

Texas A&M AgriLife Research and Extension at Dallas Center, Dallas, TX 75252, USA
* Correspondence: genhua.niu@ag.tamu.edu
† Former Senior Research Associate.

**Abstract:** This study was carried out to determine the effect of three commercial LEDs of different spectra with or without far red (FR) photons on the growth, morphology, and mineral content of two leafy vegetables under two temperatures (30 °C and 21 °C). The two leafy greens were 'Cegolaine' lettuce (*Lactuca sativa*) and 'Petite Star' pak choy (*Brassica rapa* subsp. *Chinensis*). In each temperature, there were three light spectra: red and blue LED, and white LED with or without FR. All spectra of lights were adjusted to a total photon flux density of 250 μmol m$^{-2}$ s$^{-1}$ at the top of the plant canopy. Results indicated that temperature treatment had a significant influence on most measured parameters. When temperature increased from 21 to 30 °C, lettuce shoot fresh and dry weights increased by 30% and 53%, respectively, while those of pak choy increased by approximately 22%. For both species, plants at high temperature had a larger leaf area but lower mineral content compared to those at low temperature. The spectrum treatment had a minor or no effect on the measured traits. In conclusion, the 5% FR did not impact the yield or biomass of either crop and the plant responses to spectra varied with temperature and species. The two temperatures resulted in significant differences in growth, morphology, and leaf mineral content in both species.

**Keywords:** far red photon; indoor farming; light interception; shoot biomass

## 1. Introduction

With the world population projected to reach 9 billion by 2050, more food needs to be produced for the growing population. Due to climate change with more frequent extreme weather and a limited supply of high-quality irrigation water and arable land in some regions, controlled environment agriculture such as indoor vertical farming is gaining popularity in densely populated urban settings. The innovative cultivation method of indoor vertical farming uses significantly less land, water, and fertilizer; minimizes the use of pesticides; and has the potential to achieve high crop productivity year-round with little influence from outside weather [1]. Furthermore, indoor farms can be built near consumers, which reduces the costs of transportation and increases the freshness and quality of the produce. Despite the above benefits, the high energy cost is still the biggest barrier to commercial vertical farming [1,2]. The electricity cost for lighting and air conditioning is a significant component of the high operational cost [3]. Co-optimizing multiple key growing conditions to increase light use efficiency and improve crop yield and quality has the potential to indirectly contribute to a reduction in operational costs [3]. To achieve this goal, understanding crop responses to key environmental conditions such as light spectrum and temperature is the first critical step in indoor farming.

Light-emitting diodes (LEDs) are increasingly being used for sole-source lighting in indoor farming due to their high lighting efficiency and capability to precisely control the light spectrum. Crop yield is largely determined by the amount of light the canopy

intercepts and by the photosynthetic efficiency at which the absorbed photons are converted into biomass [4,5]. Thus, light spectra that induce leaf expansion can increase light interception. Recent studies in indoor farms have shown that including far red (FR) photons with photosynthetic photons (400 to 700 nm) can improve crop yield by increasing leaf expansion and thus light interception due to shade-avoidance syndrome, as well as enhancing photosynthesis, which is known as Emerson enhancement effect [6]. For example, adding FR photons to red/blue or warm-white light synergistically increases the quantum yield of photosystem II (PSII) and the leaf net photosynthetic rate of lettuce plants [7]. The Emerson enhancement effect was also confirmed at a whole-plant-canopy level in 14 diverse species [4]. Furthermore, supplementing FR photons during the day (photoperiod) or at the end of the photoperiod for 1 h per day increased lettuce biomass by 39% and 25%, respectively, and increased lettuce leaf area by 27% to 49% [8]. However, the effect of FR photons on plant morphology and biomass of lettuce plants was cultivar-specific [9]. When lettuce plants were grown under warm-white LED light at 204 $\mu$mol m$^{-2}$ s$^{-1}$, adding FR photons up to 75.4 $\mu$mol m$^{-2}$ s$^{-1}$ increased the shoot dry weights of 'Cherokee' and 'Little Gem' lettuce by 39.4% and 19.0%, respectively, but not that of 'Green SaladBowl'.

Nevertheless, most relevant studies had an FR portion of at least 14% or higher in the total photon flux density (TPFD) [7,10]. The price of an LED light fixture with a high FR portion is often higher than those with no or a small FR portion. This is because the LED chips for the FR waveband have a much smaller application area, which is primarily used in signage, indicators, and now horticulture lighting. On the other hand, LED chips with a peak wavelength between 360 nm to 550 nm are used for general illumination and manufactured at a much larger scale; thus, the unit price is lower [11–13]. Moreover, the photosynthetic photon efficacy (PPE, $\mu$moles of photons per joule) of a horticulture LED fixture is calculated as photosynthetic photons (400–700 nm) divided by energy use, hence the PPE of fixture with FR photons is lower than that without FR photons. Because of this reason, increasing the percentage of FR in TPFD is less attractive among commercial LED manufacturers and distributors. Thus, it is necessary to determine the minimum portion of FR photons in TPFD that can improve yield. Liu and van Iersel (2022) [9] conducted a study to determine the optimal amount of supplemental FR of three lettuce cultivars, and their findings showed that the FR effect was cultivar-specific and the biomass of two cultivars was linearly correlated with FR photons in the range of 0 to 75.4 $\mu$mol m$^{-2}$ s$^{-1}$. However, their base light, the warm-white LED, already had FR photons of 5.3 $\mu$mol m$^{-2}$ s$^{-1}$.

Recent research indicates that FR and temperature interactively influence growth and morphology in 'Rex' lettuce (*Lactuca sativa*) and 'Genovese' basil (*Ocimum basilicum*) plants [14]. Specifically, as the percentage of FR photons increased from 0 to 20%, leaf length and plant height increased in both lettuce and basil; however, the magnitude of those FR-induced morphological changes generally diminished under warmer temperatures. Similarly, leaf area and biomass increased when the percentage of FR photons increased at cooler temperatures; however, under warmer temperatures (28/28 °C, day/night), growth of the two species did not change, or even decreased.

For indoor farming, in addition to narrow-band red and blue (RB) LEDs, "full-spectrum" white LEDs with or without a small percentage of FR are most commonly used for leafy green production. Considering all above factors, the objectives of this study were to quantify the effect of three representative commercial LEDs with different spectra on the growth, morphology, and leaf mineral nutrition of two leafy greens under two temperatures.

## 2. Materials and Methods

### 2.1. Plant Materials and Culture

Seeds of lettuce (*Lactuca sativa*, 'Cegolaine', Osborne quality seeds, Mount Vernon, WA, USA) and pak choy (*Brassica rapa* subsp. *Chinensis*, 'Petite Star'; Kitazawa Seed Co., Salt Lake City, UT, USA) were sown in 72-cell plug trays filled with Sunshine #1 potting mix (Sun Gro Horticulture, Agawam, MA, USA) on 18 April and 27 May 2022, for trials

1 and 2, respectively. Seedlings with three true leaves of both lettuce and pak choy were transplanted on 29 April and 7 June 2022, for trials 1 and 2, respectively, to square plastic pots (9 by 9 cm; 500 mL) filled with the same potting mix and placed under different light and temperature treatments in growth chambers for three weeks until harvest. Plants were irrigated as needed, alternately with tap water (EC = 0.4 dS m$^{-1}$, pH = 7.8) or nutrient solution (150.0 mg L$^{-1}$ N; EC $\approx$1.4 dS m$^{-1}$, pH = 6.3); that is, plants were irrigated with nutrient solution one time and with tap water the next time. The season for irrigating the plants with tap water was to prevent salt accumulation, because plants were subirrigated without leaching. The nutrient solution was made by adding water-soluble fertilizer, Peters Professional Peat Lite Special 20-10-20 (N-P$_2$O$_5$-K$_2$O) (Jr. Peters, Allentown, PA, USA), to tap water. Specifically, the fertilizer contains ammoniacal nitrogen, 8.1%; nitrate nitrogen, 11.9%; phosphate (P$_2$O$_5$), 10%; soluble potash (K$_2$O), 20%; magnesium (Mg), 015%; boron (B), 0.025%; copper, 0.025%, chelated iron, 0.1%; chelated manganese, 0.05%; molybdenum, 0.01%; and zinc, 0.05%.

### 2.2. Experiment Design and Treatments

The experiments were carried out in two growth chambers (each with multishelf growth racks equipped with different-spectrum light for each shelf) at the Texas A&M AgriLife Research Center in Dallas, TX, USA from April to June of 2022. Nine plants were placed in a nursery tray with dimensions of 25 cm by 51 cm, corresponding to a planting density of 70 plants/m$^2$. The experiment was a two-factor factorial (two temperature treatments × three light treatments) split-plot design, and was repeated over time from 29 April to 17 May and 7 to 24 June 2022. The temperature treatments were arranged as main plots (two growth chambers), and light treatments as subplots (three shelves of growth racks in each chamber). The actual temperatures were 30.1 ± 0.5 °C (mean ± standard error) for high-temperature (HT) treatment and 21.5 ± 0.4 °C for low-temperature (LT) treatment in the first replication. In the second replication, the temperatures were 29.5 ± 0.5 °C and 22.0 ± 0.5 °C, respectively, for the HT and LT chambers. In each temperature treatment chamber, three types of GE Arize$^{\text{TM}}$ Life$^2$ LEDs with different spectra were used (provided by Hort Americas, Bedford, TX, USA): (1) blue and red combination (PPR): blue 13%, red 87%; (2) PKR: blue 8%, green, 15%, red 77%; (3) PKF: blue, 8%, green 15%, red 72%, FR 5%. The spectral distributions are shown in Figure 1.

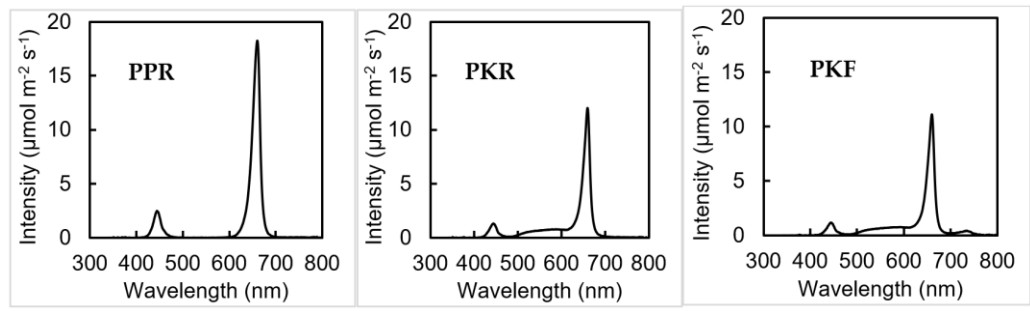

**Figure 1.** Spectral distribution of three LED spectra: PPR (blue 13%, red 87%), PKR (blue 8%, green, 15%, red 77%), and PKF (blue, 8%, green 15%, red 72%, FR 5%).

For all light spectrum treatments, the TPFD at the top of the plants was adjusted to 250 µmol m$^{-2}$ s$^{-1}$ by using a dimmer for PKR and PKF treatments to match the TPFD in PPR treatment, which was 250 µmol m$^{-2}$ s$^{-1}$ with four red and blue LED tubes with a length of 1.22 m. The growing areas for all treatments were the same: 0.6 m wide and 1.22 m long. A photoperiod of 16 h d$^{-1}$ (8:00–24:00) was applied to all treatments. The TPFD and spectrum distributions were verified and measured using a Blue Wave spectroradiometer (VIS-25; StellarNet, Tampa, FL, USA). For each species, there were a total of 9 plants (pots) per treatment.

### 2.3. Data Collection

To quantify growth and morphology as well as plant height and width, 9 plants per species per treatment were measured at harvest using a ruler. Plant height was measured vertically from pot substrate surface to the highest point of plant. Plant width was determined as the average of the two perpendicular widths. The same 9 plants were sampled for the measurement of relative chlorophyll content (SPAD index) and final biomass. SPAD was measured on the youngest fully expanded leaf using a handheld SPAD-502Plus meter (Konica Minolta, Osaka, Japan). The largest leaf was selected from each sampled plant to measure leaf length and width for both species. Fresh shoot weight was recorded right after severing the shoot from the root. After taking off all leaves from the short stem and measuring the leaf and stem weights for each plant, total leaf area (TLA) of each sampled plant was determined using the LI-3100C Leaf Area Meter (LI-COR, Lincoln, NE, USA). Dry weights of leaves and shoot (including both leaves and short stem) were determined after placing the plant tissue in a drying oven (Thermo Fisher Scientific, Waltham, MA, USA) at 70 °C until constant weight (about 4 days). Leaf/shoot water content was calculated as follows: Leaf/shoot water content (%) = (Leaf/shoot fresh weight − Leaf/shoot dry weight)/Leaf/shoot fresh weight × 100%.

To determine leaf mineral nutrition, four dry samples were ground in a Wiley mill (Thomas Scientific, Swedesboro, NH, USA) to pass a 20-mesh screen. The leaf tissues were sent to Soil Test Laboratory at Texas A&M University in College Station to analyze the following elements: nitrogen (N), phosphorus (P), potassium (K), calcium (Ca), magnesium (Mg), and sulfur (S); and micronutrients of sodium (Na), zinc (Zn), iron (Fe), copper (Cu), manganese (Mn), and boron (B). The above mineral contents were analyzed with inductively coupled plasma mass spectrometry (ICP-MS) using the methods described by Havlin and Soltanpour [15] and Isaac and Johnson [16].

### 2.4. Statistical Analysis

For each species, a two-way analysis of variance (ANOVA) was used to determine the effects of temperature (T), light spectrum (L), and T × L interaction on all measured parameters. The two replications over time were treated as two blocks. Since temperature significantly affected all growth and morphological parameters, multiple comparison was conducted separately for the two temperature levels to test the effect of light spectrum on each parameter using Tukey's honest significant difference (HSD) test at $p = 0.05$. All data were analyzed using SAS 9.4 (SAS, Cary, NC, USA).

## 3. Results

### 3.1. Growth and Morphology

Results of the analysis of variance (ANOVA) showed that for lettuce, temperature influenced all growth and morphological traits, while light spectrum only affected SPAD index, leaf area, and leaf length (Table 1). The interactive effects of temperature and light spectrum were only observed on leaf area and leaf width. The responses of pak choy to temperature and light treatments were different from those of lettuce. Temperature influenced all growth and morphological traits except for leaf and shoot water content and SPAD index. Light spectrum treatment affected leaf and shoot water content only, but not other growth and morphological traits. There were no interactions between temperature and light spectrum for any measured parameters in pak choy (Table 1).

**Table 1.** Summary of analysis of variance (ANOVA) on the effects of temperature (T) and light spectrum (L) on leaf SPAD index, plant growth, and morphology: shoot fresh weight (FW), leaf FW, total leaf area (TLA), leaf dry weight (DW), leaf water content (WC), shoot WC, leaf length (LL), and leaf width (LW) of 'Cegolaine' lettuce (*Lactuca sativa*) and 'Petite Star' pak choy (*Brassica rapa* subsp. *Chinensis*).

| Factor | SPAD | Shoot FW | Leaf FW | TLA | Leaf DW | Shoot DW | Leaf WC | Shoot WC | LL | LW |
|---|---|---|---|---|---|---|---|---|---|---|
| | | | | Lettuce | | | | | | |
| Temp. (T) | *** | *** | *** | *** | *** | *** | *** | *** | *** | *** |
| Light (L) | * | NS | NS | * | NS | NS | * | * | *** | NS |
| T ×L | NS | NS | NS | * | NS | NS | NS | NS | NS | ** |
| | | | | Pak choy | | | | | | |
| Temp. (T) | NS | *** | *** | *** | ** | ** | NS | NS | ** | ** |
| Light (L) | NS | NS | NS | NS | NS | NS | *** | *** | NS | NS |
| T × L | NS | NS | NS | NS | NS | NS | NS | NS | NS | NS |

Note: NS or *, **, and *** indicate that the treatment effect is nonsignificant or significant at a level of $p \leq 0.05$, 0.01, and 0.001, respectively.

For lettuce, compared to low-temperature treatment, the high-temperature treatment increased shoot fresh weight by 30%, dry weight by 52%, and leaf area by 64%; however, high-temperature treatment decreased leaf water content (Figure 2), and SPAD index was reduced by 7% (Supplementary Figure S1), regardless of light spectrum. High temperature also increased leaf length and width by 34% and 16%, respectively, compared to those in the low-temperature treatment. Light spectrum affected leaf water content and leaf area when plants were grown under low temperature. PKF, which is the white LED including 5% FR, increased leaf water content by 5% and leaf area by 24% compared to the PPR, which is the RB LED combination. There were no significant differences in both leaf water content and leaf area between PKF and PKR (i.e., white LED without FR) or between PKR and PPR treatments. Leaf width was about 5% smaller in PKF than PKR and PPR at high temperature. In contrast, at low temperature, leaf width was about 7% greater in PKF than PKR and PPR. At high temperature, leaf length was greater in PKF and PKR by approximately 9% compared to that in PPR; however, no significant difference was observed at low-temperature treatment.

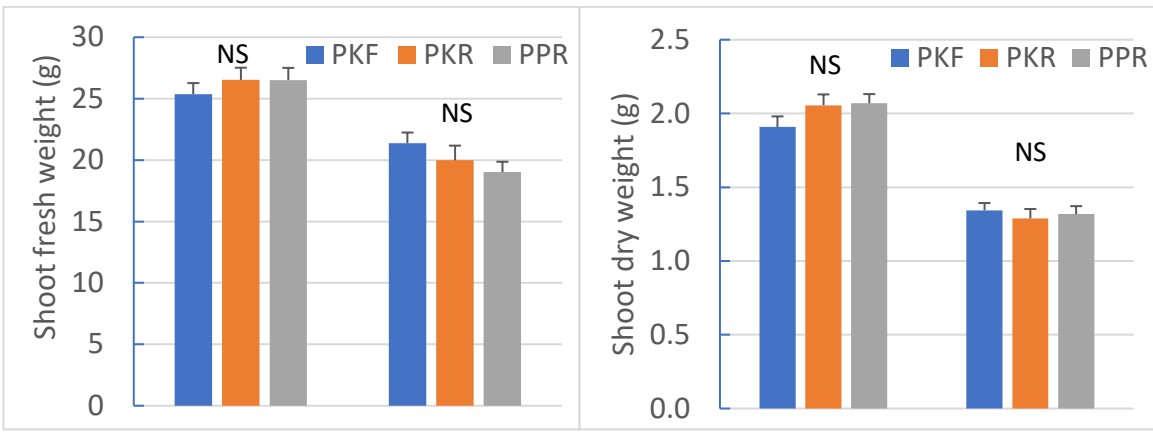

**Figure 2.** *Cont*.

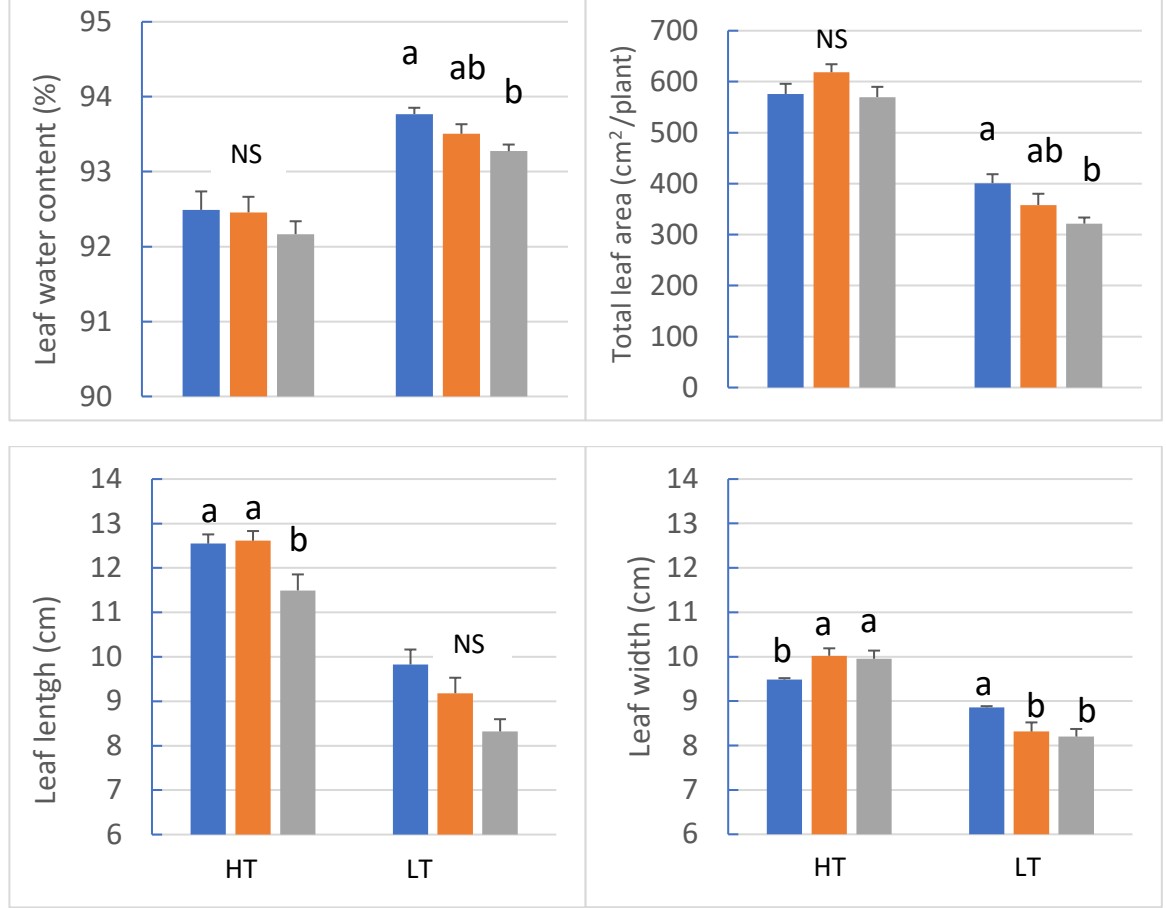

**Figure 2.** Shoot fresh weight, dry weight, leaf water content, leaf area, and leaf length and width of 'Cegolaine' lettuce (*Lactuca sativa*) plants grown under three light spectra (PKF, PKR, PPR) and two temperatures: high temperature (HT, ≈30 °C) and low temperature (LT, ≈21 °C). Vertical bars represent standard errors. Different letters for the same temperature level indicate significant differences among three light spectra at $p \leq 0.05$. NS indicates no significant differences among the light spectra.

For pak choy, high temperature increased the shoot fresh weight, shoot dry weight, and leaf area by 28%, 20%, and 38%, respectively (Figure 3). Moreover, high temperature increased leaf length by 7% but decreased leaf width by 5%. At both temperature levels, plants under PKF had higher leaf water content compared to those under PPR. Under high temperature, leaf water content was higher in PKF and PKR than that in PPR, while leaf water content was higher in PKF than those in PKR and PPR under lower temperature. The effect of light spectrum on shoot water content had a similar trend (data not presented in Figure 3). Light spectrum did not affect any other growth and morphological parameters, regardless of temperature.

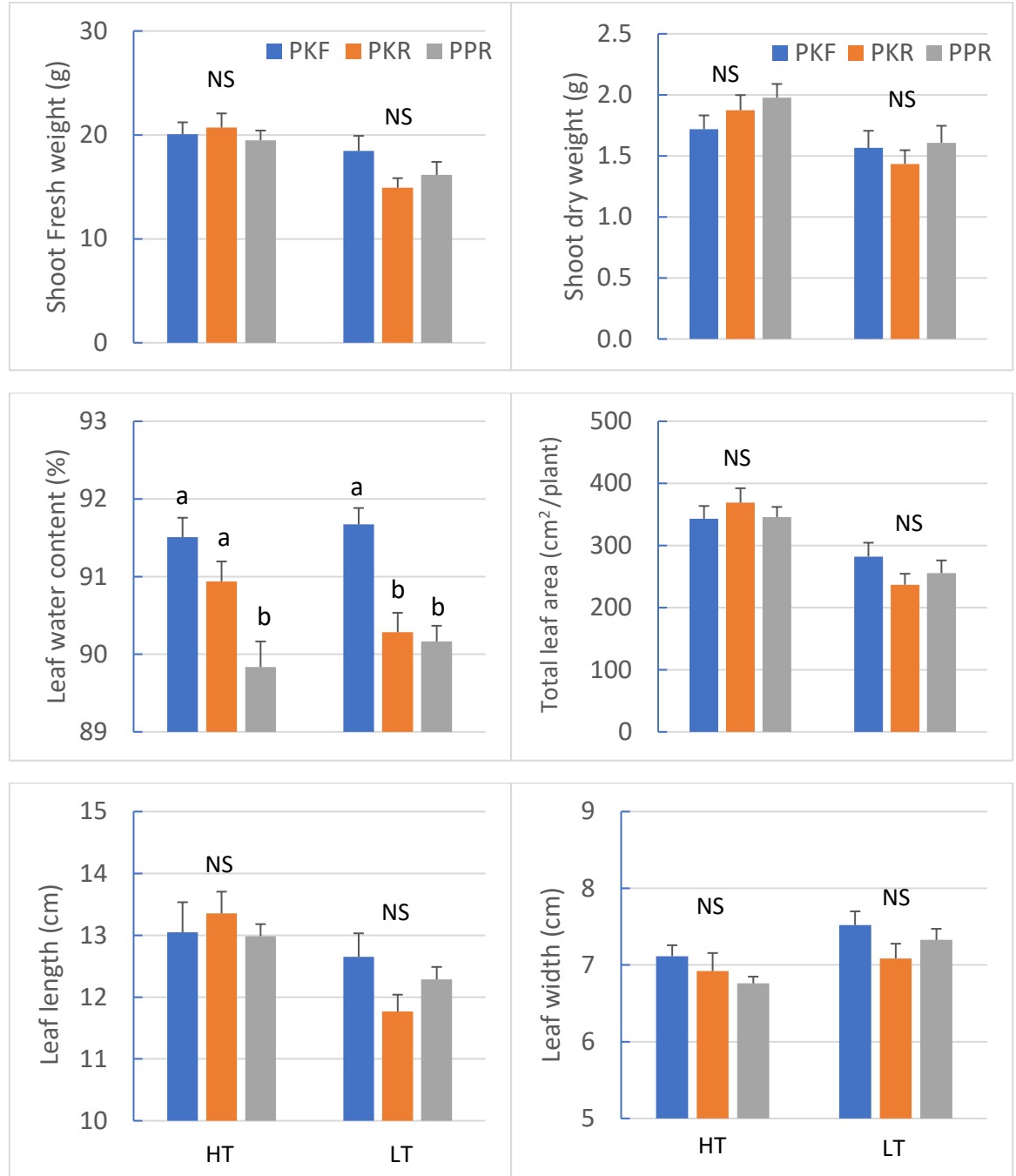

**Figure 3.** Shoot fresh weight, dry weight, leaf water content, leaf area, and leaf length and width of 'Petite Star' pak choy (*Brassica rapa* subsp. *Chinensis*) plants grown under three light spectra (PKF, PKR, PPR) and two temperatures: high temperature (HT, ≈30 °C) and low temperature (LT, ≈21 °C). Vertical bars represent standard errors. Different letters for the same temperature level indicate significant differences among three light spectra at $p \leq 0.05$. NS indicates no significant differences among the light spectra.

### 3.2. Mineral Nutrition

For leaf minerals in lettuce plants, temperature affected all the elements except for Ca, while light spectrum only affected the concentrations of N, Mg, and Cu (Table 2). The interactive effect of temperature and light spectrum was only observed in Mg and Mn. Except for S and Na, high temperature reduced the concentration of all other elements in the leaves by a range of 8% to 34%. Macronutrients N, P, K, and Mg were reduced by 18,

20, 9, and 24%, respectively. Micronutrients Zn, Fe, Cu, Mn, and B were reduced by 24, 13, 21, 19, and 34%, respectively. At high temperature, light spectrum did not affect any mineral element concentration in lettuce. At lower temperature, Mg concentration was 12% lower under PPR compared to that in the other two spectrum treatments; Cu was 14% lower in PKR than that in PKF and PPR; and B was 24% higher in PPR than that in PKF and PKR. There were some slight differences in Na and Mn among the three spectrum treatments. However, the magnitude of the light spectrum effect was smaller compared to that of temperature. Furthermore, the trend of spectrum effects on the affected mineral elements (Mg, Cu, Na, and Mn) was not clear.

**Table 2.** Macronutrients of nitrogen (N), phosphorus (P), potassium (K), calcium (Ca), magnesium (Mg), and sulfur (S); and micronutrients of sodium (Na), zinc (Zn), iron (Fe), copper (Cu), manganese (Mn), and boron (B) of 'Cegolaine' lettuce (*Lactuca sativa*) plants grown under two temperatures: high and low (HT, LT) and three light spectra: PKF (blue, 8%, green 15%, red 72%, FR 5%), PKR (blue 8%, green, 15%, red 77%), and PPR (blue 13%, red 87%).

| Factor | N | P | K | Ca | Mg | S | Na | Zn | Fe | Cu | Mn | B |
|---|---|---|---|---|---|---|---|---|---|---|---|---|
| | (%) | (g/kg) | (g/kg) | (g/kg) | (g/kg) | (g/kg) | (g/kg) | (mg/kg) | (mg/kg) | (mg/kg) | (mg/kg) | (mg/kg) |
| | | | | | High temperature (HT) | | | | | | | |
| PKF | 2.85 | 3.72 | 21.68 | 14.64 | 3.18 | 5.27 | 3.87 | 30.22 | 30.21 | 2.37 | 15.29 | 18.59 |
| PKR | 2.71 | 3.65 | 20.48 | 15.10 | 3.06 | 5.01 | 3.67 | 27.52 | 29.58 | 2.02 | 17.85 | 18.28 |
| PPR | 2.58 | 3.52 | 19.84 | 14.64 | 3.03 | 4.71 | 2.84 | 25.91 | 28.29 | 1.92 | 18.35 | 15.34 |
| | | | | | Low temperature (LT) | | | | | | | |
| PKF | 3.59 a [z] | 4.71 | 24.93 | 17.48 | 4.37 a | 4.53 a | 4.38 a | 37.24 | 34.02 | 2.61 a | 20.82 a | 25.72 b |
| PKR | 3.24 a | 4.27 | 21.16 | 15.51 | 4.13 a | 4.23 b | 2.85 b | 35.83 | 33.02 | 2.39 b | 18.32 b | 23.30 b |
| PPR | 3.05 b | 4.59 | 22.19 | 15.04 | 3.75 b | 4.31 ab | 2.97 ab | 37.21 | 33.87 | 2.94 a | 18.61 ab | 30.36 a |
| | | | | | ANOVA Summary | | | | | | | |
| T [y] | *** | *** | ** | NS | *** | *** | *** | *** | *** | *** | *** | *** |
| L | * | NS | NS | NS | ** | NS | NS | NS | NS | * | NS | NS |
| T x L | NS | NS | NS | NS | ** | NS | NS | NS | NS | NS | ** | NS |

Note: NS or *, **, and *** indicate that the treatment effect is nonsignificant or significant at a level of $p \leq 0.05$, 0.01, and 0.001, respectively. [z] different letters indicate significant differences among the three spectra at the same temperature; for means without any letters, there was no significant difference among the three spectra. [y] T: temperature; L: light spectrum.

For pak choy, the responses of leaf mineral contents to temperature and light spectrum were different from those in lettuce (Table 3). Temperature did not affect N, K, Fe, Cu, Mn, or B. Compared to low temperature, high temperature decreased the concentration of macronutrients P, Ca, and Mg by 10%, 15%, and 9%, respectively, and micronutrients Na and Zn by 39% and 22%, respectively. Similar to lettuce, leaf S was increased by 25% by high temperature relative to low temperature. Light spectrum only affected K and Na in pak choy. The spectrum effect on K was relatively small, and only observed under low-temperature conditions. Leaf Na was the lowest in PPR among all spectrum treatments at high temperature, while in PKR at low temperature a varying trend was shown in response to spectrum. An interactive effect between light spectrum and temperature was observed in Ca, Mg, Na, Fe, Cu, Mn, and B.

**Table 3.** Macronutrients of nitrogen (N), phosphorus (P), potassium (K), calcium (Ca), magnesium (Mg), and sulfur (S); and micronutrients of sodium (Na), zinc (Zn), iron (Fe), copper (Cu), manganese (Mn), and boron (B) of 'Petite Star' pak choy (*Brassica rapa* subsp. *Chinensis*) plants grown under two temperatures: high and low (HT, LT) and three light spectra: PKF (blue, 8%, green 15%, red 72%, FR 5%), PKR (blue 8%, green, 15%, red 77%), and PPR (blue 13%, red 87%).

| Factor | N | P | K | Ca | Mg | S | Na | Zn | Fe | Cu | Mn | B |
|--------|------|------|------|------|------|------|------|------|------|------|------|------|
| | (%) | (g/kg) | (g/kg) | (g/kg) | (g/kg) | (g/kg) | (g/kg) | (mg/kg) | (mg/kg) | (mg/kg) | (mg/kg) | (mg/kg) |
| | | | | | High temperature (HT) | | | | | | | |
| PKF | 2.93 | 3.90 | 21.23 | 17.20 | 2.81 | 8.87 | 2.66 a | 33.61 | 32.72 | 2.59 a | 17.99 b | 27.56 |
| PKR | 2.81 | 3.94 | 20.01 | 18.17 | 2.70 | 8.43 | 2.32 a | 31.41 | 32.14 | 2.07 ab | 23.61 a | 27.52 |
| PPR | 2.57 | 3.65 | 18.26 | 16.79 | 2.55 | 7.80 | 1.19 b | 29.59 | 28.34 | 1.87 b | 22.43 a | 22.39 |
| | | | | | Low temperature (LT) | | | | | | | |
| PKF | 3.27 a [z] | 4.49 | 21.55 a | 23.01 a | 3.03 | 6.90 | 4.89 a | 41.48 | 30.09 ab | 1.82 | 23.90 a | 22.97 b |
| PKR | 2.75 b | 3.93 | 17.65 b | 19.11 b | 2.77 | 6.41 | 2.08 b | 37.46 | 26.63 b | 1.98 | 17.61 b | 19.80 b |
| PPR | 2.88 ab | 4.37 | 19.25 ab | 19.27 ab | 3.10 | 6.82 | 3.20 ab | 42.58 | 32.61 a | 2.39 | 21.84 a | 33.79 a |
| | | | | | ANOVA summary | | | | | | | |
| T [y] | NS | ** | NS | *** | *** | *** | *** | *** | NS | NS | NS | NS |
| L | NS | NS | * | NS | NS | NS | *** | NS | NS | NS | NS | NS |
| T x L | NS | NS | NS | * | * | NS | ** | NS | ** | ** | *** | *** |

Note: NS or *, **, and *** indicate that the treatment effect is nonsignificant or significant at a level of $p \leq 0.05$, 0.01, and 0.001, respectively. [z] different letters indicate significant differences among the three spectra at the same temperature. [y] T: temperature; L: light spectrum.

## 4. Discussion

### 4.1. Light Spectrum Does Not Affect Plant Biomass Regardless of Crop Species and Temperature

In recent years, several studies have shown that FR photons synergistically interact with shorter-wavelength photons to increase leaf photochemical efficiency, leaf area, and ultimately biomass [4,10,17]. However, the percentage of FR photons in TPFD in these studies was at least 14% or higher. A high percentage of FR photons usually increases the price of a LED fixture. Thus, a minimum amount of FR that does not significantly increase the LED fixture price is desirable. In the current study, three representatives of commercial LEDs: blue and red combination (PPR), and full spectrum with 5% FR (PKF) or without (PKR) were compared. The FR percentage in PKF treatment was only 5%, replacing an equal amount of red photons, and all treatments had the same TPFD. We did not see any difference in biomass among the three spectra, regardless of crop species or temperature. The results in this study indicate that the 5% substitution of FR for red photons was not sufficient to show the Emerson enhancement effect in terms of biomass. In other words, the 5% FR did not exhibit a synergistical effect in increasing photosynthetic efficiency, as reported previously when a high percentage of FR photons was included [17].

Nevertheless, many recent studies on various crops reported increased plant growth and biomass by inclusion of FR photons to sole-source lighting in indoor farming and in greenhouses. As an example, adding FR photons at 10% to 40% of TPFD to a background of shorter-wavelength photons increased canopy photosynthesis in 14 diverse species, and the added FR photons were as effective for canopy photosynthesis as the background white photons [4]. Supplementing FR photons at 14% of TPFD to blue (B) and red (R) photons increased leaf length and shoot weight of 'Rex' and 'Cherokee' lettuce and 'Genovese' basil seedlings with more pronounced impacts under high B:R than low B:R [10]. However, addition of FR reduced relative specific chlorophyll content, although FR effects were attenuated under the high PPFD. Similar positive effects such as increased leaf area, canopy size, and/or enhanced photosynthesis efficiency of adding FR photons have been reported by other researchers [18–20]. On the other hand, negative effects have also been reported, such as decreased pigmentation and concentrations of phytochemicals when FR photons were included [20,21].

*4.2. Light Spectrum Effects on Plant Morphology and Mineral Content Are Species- and Temperature-Dependent*

For lettuce morphology, the 5% FR inclusion (PKF) increased leaf area compared to that in the PPR (red and blue) spectrum under a lower temperature, while no statistical differences were found between PKF and PKR. However, this positive effect on leaf area disappeared under high temperature, indicating an interaction between FR and temperature. Jeong et al. [14] found a similar interaction in lettuce 'Rex' with 10% and 20% FR. This may be due to thermal reversion, which is the process of light-independent but temperature-regulated phytochrome (Pfr) relaxation [22]. The phytochromes Pr (red light-absorbing and biologically inactive) and Pfr (FR-absorbing and biologically active) are interconvertible upon absorption of R (red) or FR photons. Pfr can relax back to the Pr form at high temperature. High temperature has a similar effect as FR on phytochromes. In this case, the FR effect can be covered by high temperature, which was observed in lettuce in this study. In pak choy, light spectrum only influenced leaf and shoot water content, where plants grown under PPR had a lower water content compared to those in PKF with 5% FR photons at high temperature level, and leaf water content under both PKR and PPR was lower than that in PKF. However, the differences were small, though statistically different (all within 90 to 92%, Figure 3). These results may indicate that light spectrum influences the water status of pak choy plants but does not affect carbon assimilation.

*4.3. Temperature Influences Biomass, Morphology, and Mineral Nutrition*

Temperature impacts a variety of physiological processes in plants, including photosynthesis, uptake of mineral elements, and morphology, as shown in this study in both species. For a given crop, there are three cardinal temperatures: minimum temperature, below which plant development stops; optimal temperature, at which development rate is maximal; and maximum temperature, above which development stops [23]. Between minimum and optimal temperature, plant growth and development rates increase linearly with temperature. For indoor farming, temperatures are usually maintained well above minimum temperature but below maximum temperature. To achieve the highest yield in an indoor farm, temperature should be kept near or above optimal temperature but below maximum temperature. There is limited information on how temperature influences the growth, development, and quality of leafy greens grown in indoor farms under electric lights. In this study, temperature treatment showed great effects on plant biomass, morphology, and mineral nutrition for both species. Increasing temperature from 21 °C to 30 °C significantly increased plant biomass, and leaf expansion, but reduced mineral nutrient concentrations. It is understandable that increased leaf expansion by high temperature can increase light interception, and thus improve photosynthesis and plant biomass. The decreased mineral nutrient concentrations at higher temperature might be related to the dilution effect of increased plant biomass. It might be possible to increase the amount of fertilizer applied to the plants at high temperature to increase the mineral content, which needs to be confirmed in future study.

Limited information is available regarding the optimal temperature for leafy greens in indoor farming to achieve the highest economic benefit. This is because the temperature in an indoor farm can significantly impact the electricity cost of air conditioning, depending on the outside conditions and thermal insulation of the structure. It is worth noting that the two species in the present study responded differently to high-temperature treatment. When temperature increased from 21 °C to 30 °C, the biomass increase was greater in lettuce (by 53%) compared to that in pak choy (by 20%). This result may indicate that the optimal temperature for lettuce is higher than that for pak choy. Since mineral nutrient concentrations were higher at low than high temperatures in both species, the optimal temperatures for both species may be lower than 30 °C. In addition to average daily temperature, different day and night temperatures can also alter plant mineral composition. For example, day and night temperatures of 15/25 °C (negative DIF) increased Ca, K, and Mg content in the fruit, root, and stem of eggplant and tomato [24]. Moreover, the

mineral content of leafy vegetables varied with growth stage, harvest time, environmental conditions, and cultural conditions [25–27]; thus, it is difficult to compare with other studies.

Air temperature in indoor farms not only influences plant growth, morphology, and leaf mineral content, but also metabolites, which are a group of phytochemical compounds. Polyphenolics and anthocyanins are greatly affected by growing conditions and are species-dependent. Kroggel et al. [28] reported seasonal changes in total phenolics and anthocyanin concentrations, which were typically lower during summer compared to those measured in winter. Lee et al. [29] report that photosynthetic capacity and ascorbic acid content of chicory leaves were higher at 25 °C than in other temperatures of 20 °C and 30 °C. In garland chrysanthemum, photosynthetic capacity was the greatest in both 20 °C and 25 °C, while ascorbic acid content was the greatest in 25 °C. These results indicated that in terms of phytochemicals, the optimal temperature for maximum concentration varies with specific compound. In indoor farming, controlling temperature regimes is relatively easy compared to greenhouse and field production. More studies are needed to determine the optimal temperature ranges, including an optimal diurnal temperature, to balance high yield and mineral nutrition and phytochemicals for high-value crops.

## 5. Conclusions

The three representative commercial LED spectra did not influence the biomass of lettuce and pak choy, possibly due to the small differences in spectra used in this study. The 5% FR photons did not influence the plant biomass accumulation of lettuce and pak choy. Lettuce and pak choy responded to light spectra slightly differently: light spectra regulated leaf morphology and SPAD index in lettuce plants, while in pak choy, it only affected shoot and leaf water content, but not morphology. At a high temperature of 30 °C, lettuce fresh and dry weights increased by 30% and 53%, respectively, while those of pak choy increased by approximately 22% compared to those at 21 °C. However, leaf mineral contents for most elements in both species were reduced under high-temperature conditions. More research is needed to determine the species or cultivar-specific optimal temperatures for maximum biomass using commonly available commercial LEDs.

**Supplementary Materials:** The following supporting information can be downloaded at: https://www.mdpi.com/article/10.3390/horticulturae9030331/s1, Figure S1: Relative chlorophyll content (SPAD index) of lettuce plants grown under three light spectrums (PKF, PKR, and PPR) and under high or low temperature (HT, 30 °C; LT, 21 °C). Three spectrums: PKF (Blue, 8%, green 15%, red 72%, FR 5%), PKR (Blue 8%, green, 15%, red 77%), and PPR (Blue 13%, red 87%). NS indicates no significant differences among the spectra at the same temperature.

**Author Contributions:** Conceptualization, G.N.; methodology, Y.K. and G.N.; investigation, Y.K.; resources, G.N.; data curation, Y.K.; writing—original draft preparation, G.N.; writing—review and editing, J.M. and Y.K.; supervision, G.N.; project administration, G.N.; funding acquisition, G.N. All authors have read and agreed to the published version of the manuscript.

**Funding:** This research was supported by Texas A&M AgriLife Research and USDA Hatch project TEX07726. No external funding was received.

**Acknowledgments:** We appreciate Hort Americas (Bedford, TX, USA) providing the LED fixtures for this study.

**Conflicts of Interest:** The authors declare no conflict of interest.

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
