# Peer review of "Temperature and Light Spectrum Differently Affect Growth, Morphology, and Leaf Mineral Content of Two Indoor-Grown Leafy Vegetables"

_horticulturae, doi:10.3390/horticulturae9030331_

Round 1

Reviewer 1 Report

The purpose of this manuscript was to determine the effect of three commercial LEDs of different spectrums with or without far red (FR) photons on the growth, morphology, and mineral content of two leafy vegetables grown under different temperatures.

There are few comments.

1.     Please put word line numbers

2.     In abstract, put ‘and’ between 30 and 21

3.     In abstract, please put yield value.

4.     What are the leaf greens? Put species names of the samples

5.     In introduction, put (  fron of Ocimum basilicum

6.     In introduction ‘warmer temperature (28/28 ℃)’, it the two temperature indicates day and night temperature?

7.     In method and material, what are the day and night time length?

8.     In figure 1, words of PRR, PKR, PKF were not right places in figures. Please fix them

9.     In figure 2, please put letters on all bar graphs, there were missing letters in figures. If there were no significant differences between bars, then put n.s. middle of all  three bars.

10.  Please, rephrase sentence of “In a warm region with high outside temperature, maintaining a high temperature in 54 an indoor farm may reduce electricity for air conditioning. It is worth noting that two 55 species in the present study responded differently to high temperature treatment”.

11.  The conclusion is too short, author should put what author learn from the study and suggest further study, not just summary of highlights in the experiment.

Reviewer 2 Report

Please, See my comments in the manuscript

- could you explain mor precisely why you selected the tested parameters

- what about Nitrate a very important for the market acceptance 

- finally would you suggest using of far red LED in avery case

Author Response

Thanks for the comments. We have revised accordingly. Please see attached the responses.

Reviewer 3 Report

Congrulations for the manuscript, it is a very important issue in horticulture.

The authors should broaden the discussion, that is, compare the results with other similar works. 

Author Response

Thanks for your comments. We have broadened the discussion. In the previous version, we simplified the citation of other studies in the discussion. In this revised version, we expanded the discussion by describing the previous studies on the same topic (far red photons/light). We emphasized the differences between ours (using commercial LEDs, one of them with 5% far red photons) and those in the literature with far red photons at minimum of 14% and up to 40%.

For detail, please see lines 276-303.
